# Brain Waste Removal System and Sleep: Photobiomodulation as an Innovative Strategy for Night Therapy of Brain Diseases

**DOI:** 10.3390/ijms24043221

**Published:** 2023-02-06

**Authors:** Oxana Semyachkina-Glushkovskaya, Ivan Fedosov, Thomas Penzel, Dongyu Li, Tingting Yu, Valeria Telnova, Elmira Kaybeleva, Elena Saranceva, Andrey Terskov, Alexander Khorovodov, Inna Blokhina, Jürgen Kurths, Dan Zhu

**Affiliations:** 1Institute of Physics, Humboldt University, Newtonstrasse 15, 12489 Berlin, Germany; 2Department of Biology, Saratov State University, Astrakhanskaya 82, 410012 Saratov, Russia; 3Sleep Medicine Center, Charité-Universitätsmedizin Berlin, Charitéplatz 1, 10117 Berlin, Germany; 4Optics Valley Laboratory, Huazhong University of Science and Technology, Wuhan 430074, China; 5Advanced Biomedical Imaging Facility, School of Optical Electronic Information, Huazhong University of Science and Technology, Wuhan 430074, China; 6Britton Chance Center for Biomedical Photonics, MoE Key Laboratory for Biomedical Photonics, Wuhan National Laboratory for Optoelectronics, Advanced Biomedical Imaging Facility, Huazhong University of Science and Technology, Wuhan 430074, China; 7Potsdam Institute for Climate Impact Research, Department of Complexity Science, Telegrafenberg A31, 14473 Potsdam, Germany

**Keywords:** brain diseases, meningeal lymphatic vessels, photobiomodulation

## Abstract

Emerging evidence suggests that an important function of the sleeping brain is the removal of wastes and toxins from the central nervous system (CNS) due to the activation of the brain waste removal system (BWRS). The meningeal lymphatic vessels (MLVs) are an important part of the BWRS. A decrease in MLV function is associated with Alzheimer’s and Parkinson’s diseases, intracranial hemorrhages, brain tumors and trauma. Since the BWRS is activated during sleep, a new idea is now being actively discussed in the scientific community: night stimulation of the BWRS might be an innovative and promising strategy for neurorehabilitation medicine. This review highlights new trends in photobiomodulation of the BWRS/MLVs during deep sleep as a breakthrough technology for the effective removal of wastes and unnecessary compounds from the brain in order to increase the neuroprotection of the CNS as well as to prevent or delay various brain diseases.

## 1. Therapeutic Properties of Activation of Brain Waste Removal System (BWRS) during Deep Sleep

What do we sleep for? A widespread belief is that sleep is a crucial function of the brain that is necessary for our recharging, leaving our energy refreshed when we wake up. Good sleep also helps to maintain our health and protects against the development of various diseases, including brain pathologies [1]. When we do not get enough sleep, our brain and body cannot function properly. Long-term sleep deficit can lead to the formation of dementia. Indeed, the results of a recent study on a large group of 8000 volunteers observed from their middle age till their 70th year of age for 25 years revealed a higher incidence of dementia in people aged 50–60 years who chronically did not get enough sleep (6 h and less a night) [2]. It is interesting to note that even one night of sleep deprivation causes an increase in the level of beta-amyloid (Aβ) protein in the brain of young and healthy volunteers [3,4]. There is a growing body of evidence suggesting that sleep disturbance is an independent risk factor of cognitive impairment, and the measurement of sleep quality can be an innovative method to screen for Alzheimer’s disease (AD) [5,6,7,8,9,10,11,12,13,14]. It is well known that people with AD experience poor and short sleep, which is associated with increased Aβ deposition in their brains [8,11]. Experimental and clinical studies have shown that the Aβ content in cerebral spinal fluid (CSF) is the highest in the evening before sleep and the lowest in the morning after sleep [3,15]. There is the hypothesis that sleep is accompanied by Aβ clearance from the brain [3,15]. It was recently discovered that sleep deficit causes the opening of the blood–brain barrier (BBB) to inflammatory mediators and immune cells in both humans and rodents [16,17,18,19,20]. Sleep is considered an important biomarker and a promising therapeutic target for cerebral small-vessel diseases, including AD and brain pathologies associated with BBB disruption [16]. Chronic sleep restriction promotes astrocytic phagocytosis of synaptic elements and microglia activation, i.e., the brain begins to “eat” itself [21]. Obviously, sleep is essential for the health of the central nervous system (CNS). However, the mechanisms underlying the phenomenon of restorative sleep remain unknown.

Aristotle came up with the brilliant idea that sleep triggers brain cleansing [22]. After 2000 years, his idea was confirmed by several studies suggesting that sleep activates unique brain processes for the removal of metabolites and wastes [15,23,24,25,26,27,28] (Figure 1). However, not all sleep is crucial to CNS health. There are two types of sleep: non-rapid eye movement (NREM) or deep sleep and rapid eye movement (REM) sleep, when we dream. Current studies demonstrate that only NREM sleep is optimal for functions of the BWRS [29,30]. BWRS activity is significantly increased during NREM sleep and is dramatically suppressed during wakefulness [15]. Animal data clearly demonstrate a 95% reduction in Aβ removal from the brain during wakefulness and the activation of this process during NREM sleep due to a 60% increase in interstitial fluid (ISF) space [15]. Later, it was confirmed that sleep induces the enhancement of BWRS activity arising from the expansion of ISF space [31]. Body posture during sleep is also important for the optimization of BWRS function [32]. Lee et al. reported that the horizontal position of the body during sleep contributes the most to optimal waste removal, including Aβ clearance from the brain [32].

The BWRS is the most efficient during slow-wave activity (SWA; 0–4 Hz), which is a major rhythm and a marker of NREM [29,30]. The physiological significance of SWA remains a mystery, but there is growing evidence that SWA plays an essential role in the control of sleep quality and the optimization of BWRS function [5,15,30,33,34,35,36,37]. SWA is strongly controlled, and sleep loss immediately causes a compensatory increase in SWA time during subsequent nocturnal sleep [33]. Interestingly, it is possible to suppress REM sleep in mice [38]. However, among the hundreds of mutant mouse models, no mice have yet been discovered that exhibit NREM sleep loss [39]. SWA is considered brain activity supporting the sleep regulation of the BWRS [23]. Results of contrast-enhanced magnetic resonance imaging (MRI) reveal that the circadian rhythm is an important driving force for the movement of wastes in ISF space and their clearance from the brain [40,41]. The deletion of aquaporin channels (AQP4) eliminates the effects of the circadian rhythm on brain fluid transport and the clearance of proteins from the brain [41]. Since AQP4 in astrocytic endfeet is controlled by the circadian rhythm, AQP4 optimizes brain fluid movement and waste clearance [29,41,42]. There is evidence that astrocytes control circadian timekeeping via glutamate signaling [43]. Thus, astrocytes and AQP4 present a checkpoint for BWRS function during NREM sleep [29,44]. These pioneering findings were obtained in rodents, whose circadian rhythm differs significantly from that of humans [45]. Presently, the exact contribution of the circadian rhythm to the optimization of processes of waste clearance from the human brain remains unknown [41]. Therefore, further studies are necessary to confirm the circadian control of the BWRS in humans. In addition, SWA is the result of metabolic and neurochemical changes in the sleeping brain that provide the predominance of specific neural network oscillations associated with changes in different physiological parameters, such as blood pressure, heart rate and respiratory changes, which can also regulate the BWRS [46,47,48,49].

Another factor that concerns SWA is temperature. During NREM sleep, the mouse brain neocortex temperature decreases by ~2 °C [50,51]. During this time, the hypothalamus stimulates the cooling of the brain, which induces NREM sleep [52]. There is the hypothesis that the cooling of the brain is a key function of sleep to preserve energy that might be linked to synaptic remodeling associated with SWA [53]. Cooling during NREM sleep stimulates the expression of the cold-inducible RNA-binding protein and RNA-binding motif protein 3 genes that are required for structural remodeling (they are also induced in some hibernators) [53]. The “cooling and cleaning” of the brain is a physiological phenomenon and an informative platform for therapy in degenerative and neuro-inflammatory diseases [1,54].

Taken together, considerable evidence suggests a functional relationship between sleep and the BWRS. The BWRS is considered a brain function related to waste and fluid transport with astrocyte-regulated mechanisms, while BWRS dysfunction is associated with different brain diseases, especially with cognitive decline [9,15,29,37,55,56,57,58,59,60,61,62,63]. Indeed, SWA is a new therapeutic target for AD [9,37]. Patients and animals with AD demonstrate SWA disturbances [11,37,64,65,66]. In humans, Aβ accumulation in the brain is correlated with both decreased SWA time and impaired memory consolidation [67,68]. SWA disruption is also reported in patients with mild cognitive impairment [69]. It is assumed that sleep impairment in cognitively normal old people could predict Aβ and tau accumulation in the brain [70]. Why SWA reflects AD pathology remains unknown. Decreased BWRS function after insufficient sleep may be related to an increase in the interstitial noradrenaline (NE) level [15], leading to a reduction in astrocytic volume [71] and the vasoconstriction of the pial arteries [72]. Aβ-mediated abnormalities in excitatory and inhibitory neurotransmission can be responsible for reducing SWA time [73]. Indeed, Aβ accumulation in the brain is accompanied by hyperactivity of the cortical neurons [73,74,75], and blocking neuron depolarization using gamma-aminobutyric acid A significantly improves NREM-SWA time in mice with AD [73]. Thus, Aβ-mediated synaptic abnormalities associated with neural hyperactivity can be another possible mechanism underlying SWA deficit in AD [73,74,75,76,77,78]. A better understanding of the astrocyte-mediated mechanisms of regulation of SWA time can open a new niche for novel pharmacological therapy in AD.

Thus, the BWRS is an important function of the sleeping brain that can be disturbed in animals and patients with different brain diseases, especially AD. Further studies of the mechanisms of sleep activation of the BWRS and injuries of these processes could offer a new understanding of the role of sleep in the etiology of neurological pathologies. These innovative findings could also contribute to the development of new technologies for the modulation of the therapeutic properties of the BWRS during deep sleep.

In general, significant progress in understanding the restorative mechanisms of sleep occurred in 2013, when the first study on the activation of the BWRS during deep sleep in mice was published [15]. The use of new optical technology, multiphoton microscopy, made it possible to study the movement of brain fluids in living mice with simultaneous EEG recording.

Later, using multiphoton microscopy, the glymphatic system was discovered, which is believed to have the function of removing metabolites and toxins and to be activated during sleep [79,80]. However, due to intensive research in this area, it has become clear that the perivascular spaces are more preferable routes for the removal of unnecessary compounds, including Aβ, from the brain than the glymphatic pathway [81]. In recent human studies, it was confirmed that deep sleep is accompanied by an increase in CSF generation and the activation of the BWRS [23]. These pioneering results are an important informative platform for the development of fundamentally new technologies for studying and controlling the restorative mechanisms of sleep [9,28,82,83]. Pilot studies on rats show that sleep and opened BBB are equally accompanied by BWRS activation, which has an identical effect on the EEG dynamics and makes it possible to identify EEG markers of increased BBB permeability [84]. Non-invasive technology for assessing the BBB using only EEG is important for monitoring BBB permeability during surgery in order to control the depth of anesthesia and for the analysis of the progression of a number of brain diseases, such as AD and Parkinson’s disease, brain trauma, diabetic and COVID-19 cerebrovascular injuries, and depression [16,85,86,87,88,89,90,91,92,93,94,95,96,97,98].

## 2. Photobiomodulation of BWRS: Innovative Strategies for Night Therapy of Brain Diseases

The meningeal lymphatic vessels (MLVs) are regarded as an important part of the BWRS [99,100,101,102,103,104,105]. BWRS dysfunction due to MLV abnormalities contributes to the development of neurodegenerative diseases, brain tumors and cerebrovascular accidents [61,62,63,102,103,104,105,106,107,108,109,110]. Emerging evidence indicates that MLVs have attracted a lot of therapeutic interest. However, currently, there are no clinically approved non-pharmacological technologies for the modulation of MLV function. Recent studies suggest that transcranial photobiomodulation (tPBM) of MLVs might be an innovative technology to target the BWRS [9,28,82,83,111,112,113,114,115].

Recent reviews highlight the actual applications of tPBM to the therapy of sleep disorders [9,116] and brain diseases associated with sleep injuries, including AD [117,118,119,120], psychological problems [121], brain trauma [122] and other neurological diseases [123,124]. tPBM, known as low-level light therapy, is a technique of non-invasive treatment with red and near-infrared light irradiation in a photo-therapeutic window (between 600 and 1300 nm) [125,126]. However, light at short wavelengths has very limited penetration into the brain due to light scattering, while light at long wavelengths has heating effects due to tissue absorption [126]. Light at wavelengths within range of 600–1300 nm has maximum penetration depth into the brain because of its minimal absorption and scattering within biological tissues; thus, it has significant PBM capability [125,126,127,128]. The light wavelength of 1300 nm has less scattering and can penetrate deeper into the brain [128]. Recent studies on mice show that tPBM (1267 nm) stimulates Aβ removal from the brain, which contributes to the improvement of their neurological status [28,129]. Li et al. demonstrated the therapeutic effects of tPBM (1267 nm) in mice in a model of intraventricular hemorrhages (IVHs) [130]. The course of tPBM accelerated red blood cell evacuation from the ventricles, which improved the neurological outcome and reduced mortality in mice [130]. tPBM (1267 nm) could also be a promising technology for the modulation of the lymphatic delivery of drugs and nanocarriers to the brain pathology bypassing the BBB [115]. These pilot findings suggest that tPBM at the light wavelength of 1267 nm could be a novel, non-invasive, readily applicable and commercially viable technology for the routine treatment of various brain diseases.

tPBM of the brain has been studied extensively for prolonged periods and numerous devices are available on the market [123]. Figure 2a illustrates a typical tPBM procedure, which includes irradiation of the patient’s head with an expanded 1064 nm laser beam [131]. Laser light radiation is applied with a handheld laser head, controlled by a physician [131,132,133], because of the potential hazard of invisible collimated laser beams and the requirement to use appropriate eye and skin protection. Light sources based on light-emitting diodes (LEDs) provide more flexible applications of tPBM, as they are safer for patients and cheaper than a laser of equivalent power density. To ensure uniform light energy distribution over a skin surface, LEDs are assembled into clusters of various sizes and constructions, rigid blocks [118,134] or flexible panels [135]. LED clusters can be fixed on a patient’s head with a headband [136], mounted inside a rigid helmet [134], or assembled in a helmet-like structure [137] or in a hat made of flexible LED panels [135] (Figure 2a–d).

To prevent the heating effect on the head, tPBM devices assembled with a large number of LEDs are used with a cooling system mounted over each LED cluster [134,137] (Figure 2b). Other therapeutic applications of PBM include the modification of ambient light with blue light emitted with an eyeglass-shaped gadget to correct delayed wake phase disorder [138] (Figure 2e).

Typically, the tPBM procedure is performed on awake patients. However, BWRS activity is much stronger during sleep than in the awake state, as we discuss above [15,28,29,30,31,32]. Based on the fact that the BWRS is activated during deep (NREM) sleep, we hypothesize that tPBM during night sleep might be more effective for the therapy of brain diseases than tPBM during the day. In our recent review, we discuss that sleep can be a promising therapeutic target for cerebral small-vessel disease [9]. In our pioneering study, we discovered that tPBM performed during sleep stimulates Aβ clearance from the mouse brain more effectively than tPBM applied during wakefulness [28]. The course of night tPBM promotes good recovery of neurological status and recognition memory in mice with AD compared with the daily course of tPBM [28]. Several reviews report that the mechanisms of therapeutic effects of tPBM during sleep could be completely different from those during the awake state [83,139]. However, the application of tPBM during sleep as a new strategy for the night therapy of brain diseases is in its infancy. Although the demand for tPBM during sleep has been clearly formulated, its practical realization has not yet been reported [83,139].

Currently, there are no technologies for simultaneous tPBM and sleep monitoring. Recent surveys of consumer sleep technologies demonstrate some demand for optional physical interventions during the sleep process aimed to improve the quality of sleep [140], although the requirement of strong scientific evidence to legitimize claims about utility, safety, and efficacy, as well as for informed choice and public trust, are emphasized [141,142]. Sleep technologies are presented on the market as various sleep-tracking devices [143,144] (Figure 3a–d).

Commercial sleep-tracking technologies include radars; near-infrared and thermal imaging devices; bedding-based sleep sensors; and wearable body sensors designed to be fixed on wrist, finger, feet, waist, chest or head [143,145]. Headbands are also capable of performing electroencephalography (EEG) recording and analysis that sufficiently improve sleep detection [142,144]. All sleep-tracking technologies use wireless interfaces to communicate with mobile devices utilizing applications for sleep monitoring.

## 3. Mechanisms of tPBM of BWRS

There are a number of studies on the mechanisms underlying tPBM of the BWRS [9,28,111,112,113,114,115,129]. It is reported that PBM causes the dilation of both MLVs and mesenteric lymphatics [114,130]. The PBM-mediated dilation of lymphatic vessels (LVs) is associated with an increase in the permeability of the lymphatic endothelium and a decrease in the expression of tight junction (TJ) proteins [114]. TJ proteins are the main components of the lymphatic endothelium and play a crucial role in the regulation of lymph movement in the lymphatic network [146]. Changes in the permeability of the lymphatic endothelium allow wastes and immune cells to be transported with lymph in LVs [147,148]. Indeed, the transport of antigens, and immune and dendritic cells across LVs is coupled with water flux and depends on the permeability of LVs [149,150]. These effects of PBM might be related to the PBM-induced production of nitric oxide (NO) in the lymphatic endothelium [82,130,151]. NO dilates blood and lymphatic vessels via the activation of soluble guanylate cyclase and protein kinase G, which stimulates the opening of calcium (Ca^2+^)-activated potassium channels [152]. The reduction in the Ca^2+^intracellular level blocks the phosphorylation of myosin light-chain kinase, leading to vascular relaxation [152]. There are several studies suggesting that PBM-mediated vascular relaxation is the result of a PBM-induced increase in the activity of endothelial NO synthase (eNOS) [82,153,154,155]. eNOS-induced NO production in the lymphatic endothelium leads to an increase in lymphatic flow and the removal of wastes and toxins from different tissues [156].

Recently, it has been shown that PBM increases lymphatic contractility and that PBM stimulates NO synthesis in isolated cells of the lymphatic endothelium [115,130]. Lymphatic contractility is important for the movement of cells and molecules in LVs and is regulated by NO [157]. These facts suggest that the PBM-induced lymphatic contractility could be the possible mechanism responsible for the PBM-mediated increase in the clearance of wastes and metabolites from the brain [112,115,130].

Another mechanism of PBM-mediated control of vascular tone is the ability of singlet oxygen to regulate endothelial relaxation [158]. PBM at the light wavelength of 1267 nm stimulates the direct generation of singlet oxygen in tissues [159,160,161,162]. Singlet oxygen induces the oxidation of the amino acid tryptophan in mammalian tissues, which leads to cell production of metabolites such as N-formylkynurenine with the activation of a heme-containing enzyme called indoleamine 2, 3-dioxygenase 1 [158]. This enzyme is widely expressed in the blood and lymphatic endothelium, contributing to the relaxation of vascular tone [158,163,164]. It was recently discovered that endothelial indoleamine 2, 3-dioxygenase 1 causes the dilation of blood vessels via the formation of singlet oxygen [158]. These pioneering findings shed light on the role of singlet oxygen in the regulation of vascular tone, opening a new perspective on the modulation of the systemic inflammatory response of the vasculature.

The PBM-mediated increase in NO production inhibits the activity of the mitochondrial cytochrome c oxidase (CCO) enzyme, leading to an increase in mitochondrial membrane potential [165,166,167]. These PBM effects contribute to an increase in the metabolic and energetic activity of cells via more oxygen consumption, more glucose metabolization and more ATP production by the mitochondria [165,166,167,168]. Recently, a mechanism of PBM-mediated stimulation of Aβ clearance from the brain via the CCO-mediated activation of the cAMP-dependent protein kinase signaling pathway was discovered [169]. The neuroprotective effects of PBM (1068 nm) due to a decrease in the Aβ level in the brain through microglia activation and angiogenesis were also demonstrated in mice with AD [170]. PBM decreases Aβ deposition in the brain by recruiting microglia to the Aβ burden [170]. PBM can reduce Aβ accumulation in the brain by increasing angiogenesis [171]. Indeed, the PBM (1070 nm)-induced increase in cerebral vessel density is positively correlated with the clearance of toxins from the brain [171]. PBM (630 nm) also improves brain drainage by contributing to Aβ removal in the APP/PS1 mouse model of AD [172].

## 4. Conclusions

In modern neurobiology, sleep is considered a novel biomarker and a promising therapeutic target for brain diseases. This is due to the recent discovery of the nighttime activation of the BRWS, which plays an important role in the removal of wastes and toxins from the brain, contributing to the neuroprotection of the CNS. The hypothesis has arisen in the scientific community that night stimulation of the BWRS might be a breakthrough strategy as a new treatment of AD, Parkinson’s disease, stroke, brain trauma and oncology. Although this research is in its infancy, there are pioneering and promising results suggesting that night tPBM stimulates the lymphatic removal of Aβ from the mouse brain more effectively than daily tPBM and is associated with a greater improvement of the neurological status and recognition memory in animals. There is the hypothesis that tPBM modulates the tone and permeability of the lymphatic endothelium by stimulating NO formation, promoting the lymphatic clearance of wastes and toxins from brain tissues (Figure 4). It is assumed that tPBM can also lead to angio- and lymphangiogenesis, which might be another mechanism responsible for tPBM-mediated stimulation of the BWRS. Typically, tPBM is used in the awake state. Therefore, there are numerous tPBM devices constructed to apply a certain dose of light irradiation without technologies for sleep monitoring. To create a technology for simultaneous tPBM and sleep monitoring, only a wirelessly controlled gadget with LED or laser sources is needed to perform tPBM during sleep. The gadget can be controlled directly using a sleep-tracking device or mobile application that integrates data from various sources, including ambient conditions, e.g., room temperature, atmospheric pressure etc. Thus, there is a crucial challenge to design autonomous LED or laser light sources that are capable of providing the required therapeutic dose of light radiation in a certain region of the patient’s head without disturbing the sleeping patient, e.g., sleep-tracking headbands [142,144]. To minimize patient discomfort, advanced materials such as flexible organic LEDs can be used [173,174].

Since the idea of tPDT of the BWRS during sleep is in its infancy, here, we only discuss animal studies, because there are no human data in this area yet. There is evidence of a possible pharmacological stimulation of lymphaneogenesis for the treatment of brain tumors and for increasing brain immunity, as well as for the lymphatic excretion of Aβ from brain tissues [175,176]. However, bringing new pharmaceuticals to the market takes 10 to 15 years. The proposed new pharmacological strategies for the stimulation of the BWRS have been tested on a small group of animals. It could take a long time for additional preclinical and clinical trials. In addition, pharmacological studies are not always successful, as in the case of the development of aducanumab, a novel anti-Aβ antibody for the treatment of AD. Two phase 3 clinical trials including 3285 participants demonstrated that aducanumab causes vasogenic edema in 35% of patients associated with different symptoms, such as headache, confusion, dizziness and nausea, and that microhemorrhages occur in 19% of AD patients [176]. Therefore, the Food and Drug Administration (FDA) required Biogen, the pharmacological company, to conduct an additional clinical trial to verify aducanumab’s clinical benefits. If the trial fails to verify clinical benefits, the FDA may initiate proceedings to withdraw the approval of aducanumab.

The purpose of our review is to highlight non-pharmacological strategies for the stimulation of the BWRS that have a high potential to be introduced into routine clinical practice, as well as those that can become smart sleep gadgets used at home and while traveling without reducing the high pace of life of modern people. In this aspect, photonic technologies are the most promising, since they are already widely used in the clinical field for the treatment of various brain diseases [117,118,119,120,121,122,123,124]. tPBM of the BWRS is the only non-pharmacological technology proposed to activate the lymphatic excretion of metabolites and toxins from brain tissues [9,28,82,83,111,112,113,114,115,129]. Further studies of optimal doses and wavelengths of tPDT, and advantages and limitations of tPBM of the BWRS in patients of different ages and with various brain diseases could significantly help in the development of guidelines for its safe use in humans. The auditory effects can affect SWA time [177], which may also influence the BWRS, and could be an interesting subject of future research.

## Figures and Tables

**Figure 1 ijms-24-03221-f001:**
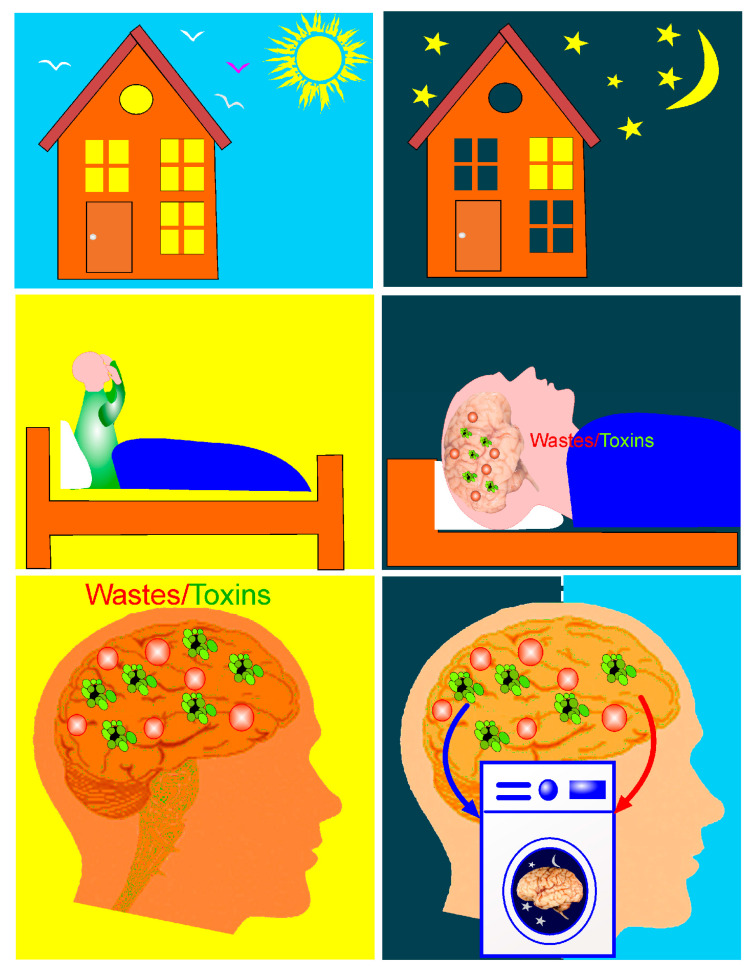
BWRS activation during sleep. During wakefulness, the BWRS is inactive; therefore, wastes and toxins accumulate in the brain. During deep sleep, the BWRS is activated, which is accompanied by an increase in the drainage of brain tissues and the intense elimination of wastes and toxins. Figuratively speaking, the brain, during deep sleep, turns into a washing machine, removing unnecessary compounds from its tissues.

**Figure 2 ijms-24-03221-f002:**
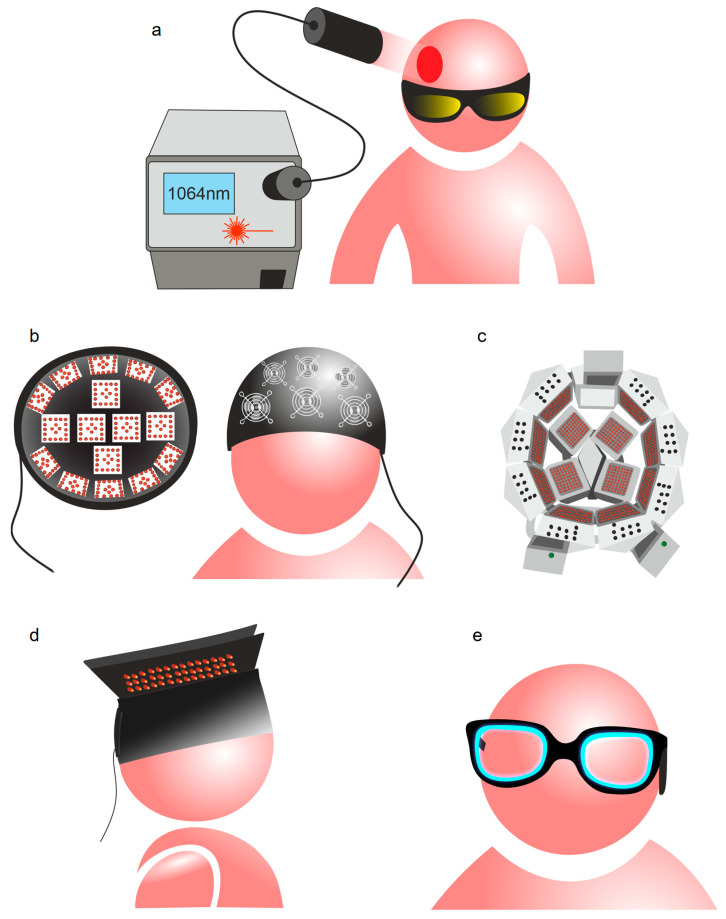
Devices for tPBM: (**a**) typical tPBM procedure includes irradiation of patient’ head with expanded 1064 nm laser beam [131], which has maximum penetration depth into the brain [125,126,127,128]; (**b**) helmet for tPBM with system of forced cooling [134]; (**c**) helmet with rigid structure assembled of 1070 nm LED clusters [137]; (**d**) hat made of flexible LED panels [135]; (**e**) smart sleep glasses [138].

**Figure 3 ijms-24-03221-f003:**
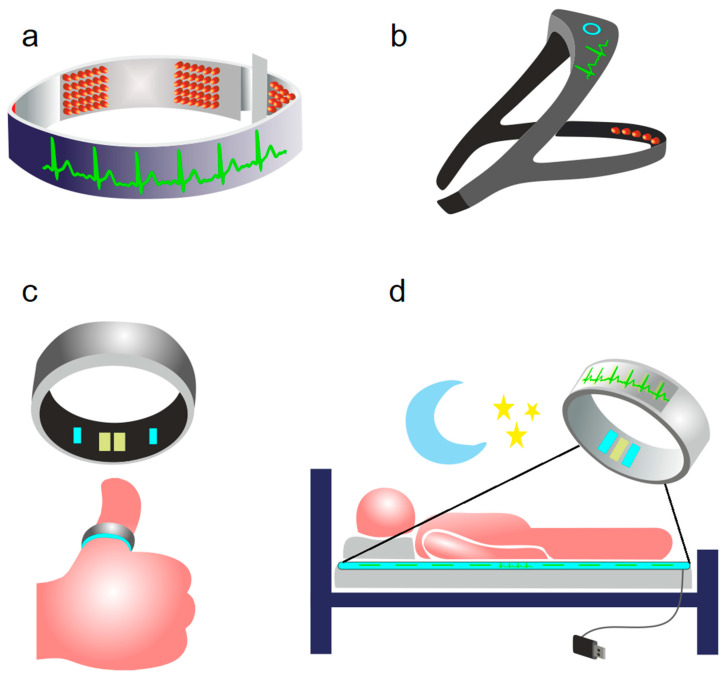
Sleep-tracking gadgets: (**a**) BrainBit headband [143]; (**b**) Philips Smart Sleep Deep Sleep Headband [142]; (**c**) Oura ring smart sleep tracker [144]; (**d**) sleep tracker 2 mm thick belt to be placed under the sheet on a bed [144].

**Figure 4 ijms-24-03221-f004:**
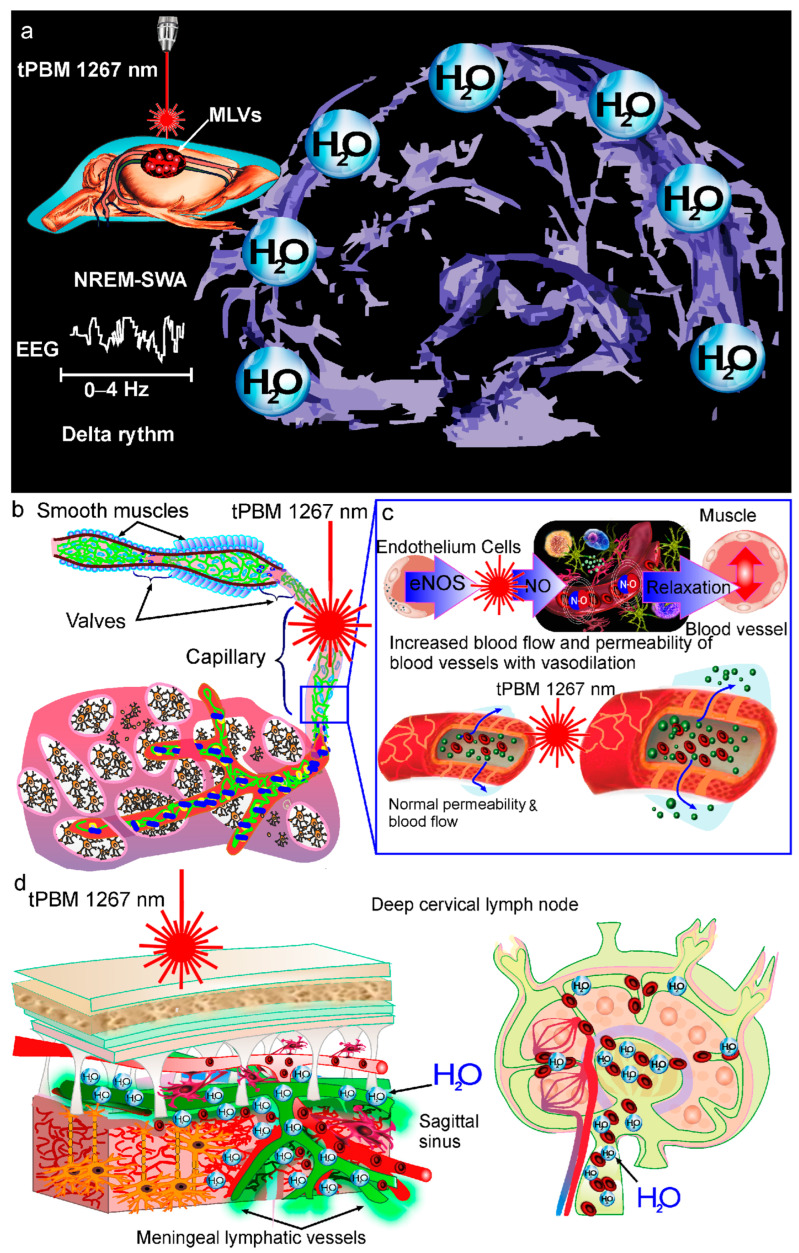
Emerging strategy of tPBM of the BRWS during deep sleep (NREM-SWA activity with delta rhythm): (**a**) Since the BRWS is activated during deep sleep, tPBM-mediated stimulation of the clearance of wastes and toxins from the brain could be more effective if tPBM were used during the NREM-SWA activity of the sleeping brain than during wakefulness [28]; (**b**–**d**) There is the hypothesis that tPBM modulates the tone and permeability of the lymphatic endothelium via the stimulation of eNOS and NO production, promoting the lymphatic clearance of wastes and toxins from brain tissues [96,137,138,139,140].

## Data Availability

Not applicable.

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
