# Peer review of "Brain Waste Removal System and Sleep: Photobiomodulation as an Innovative Strategy for Night Therapy of Brain Diseases"

_ijms, 2023, doi:10.3390/ijms24043221_

Round 1

Reviewer 1 Report

1.      Please describe the similarities and differences between this study and related past studies in the introduction section.

2.      What are the research limitations of this study? Please provide a description.

Author Response

Comments: Please describe the similarities and differences between this study and related past studies in the introduction section. What are the research limitations of this study? Please provide a description.

Response: The authors would like to express their sincere gratitude to the referee for the recommendations and great help in improving our paper. We added a summary explaining how key previous research has provided the basis for the development of fundamentally new technologies for studying and controlling the restorative mechanisms of sleep, presented in our review (Lines 143-163). We also added description of limitations of our review (Lines 333-371). All corrections are highlighted in yellow in the text of the article.

Lines 143-163:

“Thus, the BWRS is an important function of sleeping brain that can be disturbed in animals and patients with different brain diseases, especially with AD. The further studies of mechanisms of sleep-activation of the BWRS and injuries of these processes will open a new understanding of the role of sleep in the etiology of neurological pathologies. These innovative findings will also contribute the development of new technologies for modulation of therapeutic properties of the BWRS during deep sleep.

In general, significant progress in understanding the restorative mechanisms of sleep occurred in 2013, when the first study of activation of the BWRS during deep sleep in mice was published [15]. The use of new optical technology, multiphoton microscopy, made it possible to study the movement of brain fluids in living mice with simultaneous EEG recording.

Later, using multiphoton microscopy, the glymphatic system was discovered, which is believed to have the function of removing metabolites and toxins, being activated during sleep [79, 80]. However, due to intensive research in this area, it has be-come clear that the perivascular spaces are more preferable routes for the removal of unnecessary compounds, including Aβ, from the brain than the glymphatic pathway [81]. In recent human studies, it has been confirmed that deep sleep is accompanied by an increase in the CSF generation and activation of the BWRS [23]. These pioneering results are an important informative platform for the development of fundamentally new technologies for studying and controlling the restorative mechanisms of sleep [9, 28, 82, 83]. Pilot studies on rats have shown that sleep and the opened BBB are equally accompanied by the BWRS activation, which has an identical effect on the EEG dynamics and makes it possible to identify the EEG markers of the increased BBB permeability [84]. Non-invasive technology for assessing the BBB using only EEG is important for monitoring the BBB permeability during surgery in order to control the depth of anesthesia, for analysis of the progression of a number of brain diseases, such as AD and Parkinson's disease, brain trauma, diabetic and COVID-19 cerebrovascular injuries, depression [16, 85-98]”.

Lines 333-371:

“Since the idea of tPDT of the BWRS during sleep is in its infancy, here we discuss only the animal studies because there are no data in this area in humans yet. There is evidence of a possible pharmacological stimulation of lymphaneogenesis for the treatment of brain tumors and increasing brain immunity, as well as for the lymphatic excretion of Aβ from the brain tissues [175, 176]. However, bringing new pharmaceuticals to market takes 10 to 15 years. The proposed new pharmacological strategies for stimulation of the BWRS have been tested on a small group of animals. It will take a long time for additional preclinical and clinical trials. In addition, pharmacological studies are not always successful. For example, the development of aducanumab, a novel anti-Aβ anti-body, for the treatment of AD. 2 phase 3 clinical trials including 3285 participants demonstrate that aducanumab causes vasogenic edema in 35% of patients associated with different symptoms, such as headache, confusion, dizziness, and nausea as well as microhemorrhages occur in 19 % of AD patients [176]. Therefore, the Food and Drug Administration (FDA) required the Biogen, pharmacological company, to conduct additional clinical trial to verify the aducanumab’s clinical benefit. If the trial fails to verify clinical benefit, the FDA may initiate proceedings to withdraw approval of aducanumab.

The purpose of our review is to highlight non-pharmacological strategies of stimulation of the BWRS that have a high potential to be introduced into routine clinical practice, as well as which can become Smart Sleep gadgets used at home and travel without reducing the high pace of life of modern people. In this aspect, photonic technologies are the most promising, since they are already widely used in the clinic for the treatment of various brain diseases [117-124]. tPBM of the BWRS is the only non-pharmacological technology proposed to activate the lymphatic excretion of metabolites and toxins from the brain tissues [9, 28, 82, 83, 111-115, 129]. Further studies of optimal doses, wavelength and time of tPDT, advantages and limitations for the treatment of patients of different ages and various brain diseases will significantly help in our understanding of the mechanisms of tPDT-related activation of the BWRS and the development of guidelines for its safe use in humans. The auditory effects can affect the SWA time [177] that may also influence on the BWRS and can be an interesting subject of future research”.

The authors would like to thank the referee again for great help in improving the quality of our article and its possible publication in International Journal of Molecular Science.

Reviewer 2 Report

Dear authors,

I found the topic of your review interesting and these are my comments.

The authors reported some information about the therapeutic properties of activation of the brain’s waste removal system (BWRS) during deep sleep and about the photobiomodulation of the BWRS as innovative strategies for night therapy. The authors also reported the procedure of irradiation therapy and the mechanisms of transcranial photobiomodulation of the BWRS and the MLVs.

The manuscript is well-organized ad well written, in my opinion, the article needs a deepening of the topics. For example, the review would be more interesting if the authors reported some case reports or some studies on the population which use photobiomodulation if they are available.

Minor revisions

Since when has this therapy been available? What effect has it had on the quality of life?

BWRS somewhere is written wrong (RWRS)

Is The William Shakespeare referment necessary? In my opinion, this reference confers a not very scientific tone to the paper

Author Response

Comments:

The manuscript is well-organized ad well written, in my opinion, the article needs a deepening of the topics. For example, the review would be more interesting if the authors reported some case reports or some studies on the population which use photobiomodulation if they are available.

Minor revisions

Since when has this therapy been available? What effect has it had on the quality of life?

BWRS somewhere is written wrong (RWRS)

Is The William Shakespeare referment necessary? In my opinion, this reference confers a not very scientific tone to the paper

Response: The authors would like to express their sincere gratitude for the positive assessment of our review and for the important recommendations for improving its quality.

We corrected grammatical and stylistic errors as well as we removed the phrase with William Shakespeare.

The technology of photostimulation of the BWRS, which we discuss in our review, is in its infancy. Therefore, we highlight only animal data because there are no human studies in this area yet. We added in the conclusion this limitation of our review as well as prospects for future clinical research (Lines 333-371).

Lines 333-371:

“Since the idea of tPDT of the BWRS during sleep is in its infancy, here we discuss only the animal studies because there are no human data in this area yet. There is evidence of a possible pharmacological stimulation of lymphaneogenesis for the treatment of brain tumors and increasing brain immunity, as well as for the lymphatic excretion of Aβ from the brain tissues [175, 176]. However, bringing new pharmaceuticals to market takes 10 to 15 years. The proposed new pharmacological strategies for stimulation of the BWRS have been tested on a small group of animals. It will take a long time for additional preclinical and clinical trials. In addition, pharmacological studies are not always successful. For example, the development of aducanumab, a novel anti-Aβ anti-body, for the treatment of AD. 2 phase 3 clinical trials including 3285 participants demonstrate that aducanumab causes vasogenic edema in 35% of patients associated with different symptoms, such as headache, confusion, dizziness, and nausea as well as microhemorrhages occur in 19 % of AD patients [176]. Therefore, the Food and Drug Administration (FDA) required the Biogen, pharmacological company, to conduct additional clinical trial to verify the aducanumab’s clinical benefit. If the trial fails to verify clinical benefit, the FDA may initiate proceedings to withdraw approval of aducanumab.

The purpose of our review is to highlight non-pharmacological strategies of stimulation of the BWRS that have a high potential to be introduced into routine clinical practice, as well as which can become Smart Sleep gadgets used at home and travel without reducing the high pace of life of modern people. In this aspect, photonic technologies are the most promising, since they are already widely used in the clinic for the treatment of various brain diseases [117-124]. tPBM of the BWRS is the only non-pharmacological technology proposed to activate the lymphatic excretion of metabolites and toxins from the brain tissues [9, 28, 82, 83, 111-115, 129]. Further studies of optimal doses, wavelength and time of tPDT, advantages and limitations for the treatment of patients of different ages and various brain diseases will significantly help in our understanding of the mechanisms of tPDT-related activation of the BWRS and the development of guidelines for its safe use in humans. The auditory effects can affect the SWA time [177] that may also influence on the BWRS and can be an interesting subject of future research”.

The authors thank the referee for an interest in our review and for the opportunity to improve its quality with the helpful advices.

Authors

Author Response

Comments: The main focus of this review is to report recent knowledge on the action of photostimulation of the head on physiological removal of brain catabolic wastes, and on its different efficacy during the wake-sleeping circadian rhythm, outlining the potential utility of light therapy to treat different neurodegenerative diseases. The authors are active researchers in the field and know well the topic. The manuscript is well written and the content is interesting. However, the style is quite enthusiastic on the potential utilities of such new technology. We would suggest that the authors should be a little more cautious, trying to hypothesize potential harmful effects of long-term photostimulations of the brain, that likely would be required to treat chronic neurodegenerative diseases in humans.

Minor points

Some grammatical and syntactic errors are disseminated throughout the manuscript.

Response: The authors would like to express their deep gratitude to the referee for the opportunity to improve the quality of our review with constructive comments and important recommendations.

We corrected grammatical and stylistic errors in our manuscript.

The technology of photostimulation of the BWRS, which we discuss in our review, is in its infancy. Therefore, we highlight only animal data because there are no human studies in this area yet. We added in the conclusion this limitation as well as prospects for future clinical research. We draw the readers' attention to further studies of optimal doses and wavelength of tPDT, advantages and limitations for tPBM of the BWRS in patients of different ages and various brain diseases, which will significantly help in the development of guidelines for its safe use in humans (Lines 333-371).

Lines 333-371:

“Since the idea of tPDT of the BWRS during sleep is in its infancy, here we discuss only the animal studies because there are no human data in this area yet. There is evidence of a possible pharmacological stimulation of lymphaneogenesis for the treatment of brain tumors and increasing brain immunity, as well as for the lymphatic excretion of Aβ from the brain tissues [175, 176]. However, bringing new pharmaceuticals to market takes 10 to 15 years. The proposed new pharmacological strategies for stimulation of the BWRS have been tested on a small group of animals. It will take a long time for additional preclinical and clinical trials. In addition, pharmacological studies are not always successful. For example, the development of aducanumab, a novel anti-Aβ anti-body, for the treatment of AD. 2 phase 3 clinical trials including 3285 participants demonstrate that aducanumab causes vasogenic edema in 35% of patients associated with different symptoms, such as headache, confusion, dizziness, and nausea as well as microhemorrhages occur in 19 % of AD patients [176]. Therefore, the Food and Drug Administration (FDA) required the Biogen, pharmacological company, to conduct additional clinical trial to verify the aducanumab’s clinical benefit. If the trial fails to verify clinical benefit, the FDA may initiate proceedings to withdraw approval of aducanumab.

The purpose of our review is to highlight non-pharmacological strategies of stimulation of the BWRS that have a high potential to be introduced into routine clinical practice, as well as which can become Smart Sleep gadgets used at home and travel without reducing the high pace of life of modern people. In this aspect, photonic technologies are the most promising, since they are already widely used in the clinic for the treatment of various brain diseases [117-124]. tPBM of the BWRS is the only non-pharmacological technology proposed to activate the lymphatic excretion of metabolites and toxins from the brain tissues [9, 28, 82, 83, 111-115, 129]. Further studies of optimal doses and wavelength of tPDT, advantages and limitations for tPBM of the BWRS in patients of different ages and various brain diseases will significantly help in the development of guidelines for its safe use in humans. The auditory effects can affect the SWA time [177] that may also influence on the BWRS and can be an interesting subject of future research”.

The authors thank the referee for great help in improving our review for its possible publication in International Journal of Molecular Science.

Authors

Round 2

Reviewer 1 Report

The manuscript is well organized. I have no more question.